# Liposomal Nanocarriers Designed for Sub-Endothelial Matrix Targeting under Vascular Flow Conditions

**DOI:** 10.3390/pharmaceutics13111816

**Published:** 2021-10-31

**Authors:** Lauren B. Grimsley, Phillip C. West, Callie D. McAdams, Charles A. Bush, Stacy S. Kirkpatrick, Joshua D. Arnold, Michael R. Buckley, Raymond A. Dieter, Michael B. Freeman, Michael M. McNally, Scott L. Stevens, Oscar H. Grandas, Deidra J. H. Mountain

**Affiliations:** Department of Surgery, University of Tennessee Graduate School of Medicine, 1924 Alcoa Highway Box U-11, Knoxville, TN 37920, USA; Lebenner13@gmail.com (L.B.G.); pwest@utmck.edu (P.C.W.); cmcadams@utmck.edu (C.D.M.); cabush@utmck.edu (C.A.B.); skirkpat@utmck.edu (S.S.K.); jarnold@utmck.edu (J.D.A.); mbuckley@utmck.edu (M.R.B.); rdieter@utmck.edu (R.A.D.III); mfreeman@utmck.edu (M.B.F.); mmcnally@utmck.edu (M.M.M.); sstevens@utmck.edu (S.L.S.); ograndas@utmck.edu (O.H.G.)

**Keywords:** liposomes, targeted drug delivery, hemodynamic flow, vascular shear stress, vascular therapeutics

## Abstract

Vascular interventions result in the disruption of the tunica intima and the exposure of sub-endothelial matrix proteins. Nanoparticles designed to bind to these exposed matrices could provide targeted drug delivery systems aimed at inhibiting dysfunctional vascular remodeling and improving intervention outcomes. Here, we present the progress in the development of targeted liposomal nanocarriers designed for preferential collagen IV binding under simulated static vascular flow conditions. PEGylated liposomes (PLPs), previously established as effective delivery systems in vascular cells types, served as non-targeting controls. Collagen-targeting liposomes (CT-PLPs) were formed by conjugating established collagen-binding peptides to modified lipid heads via click chemistry (CTL), and inserting them at varying mol% either at the time of PLP assembly or via micellar transfer. All groups included fluorescently labeled lipid species for imaging and quantification. Liposomes were exposed to collagen IV matrices statically or via hemodynamic flow, and binding was measured via fluorometric analyses. CT-PLPs formed with 5 mol% CTL at the time of assembly demonstrated the highest binding affinity to collagen IV under static conditions, while maintaining a nanoparticle characterization profile of ~50 nm size and a homogeneity polydispersity index (PDI) of ~0.2 favorable for clinical translation. When liposomes were exposed to collagen matrices within a pressurized flow system, empirically defined CT-PLPs demonstrated significant binding at shear stresses mimetic of physiological through pathological conditions in both the venous and arterial architectures. Furthermore, when human saphenous vein explants were perfused with liposomes within a closed bioreactor system, CT-PLPs demonstrated significant ex vivo binding to diseased vascular tissue. Ongoing studies aim to further develop CT-PLPs for controlled targeting in a rodent model of vascular injury. The CT-PLP nanocarriers established here show promise as the framework for a spatially controlled delivery platform for future application in targeted vascular therapeutics.

## 1. Introduction

Endovascular interventions are commonly used to treat peripheral vascular disease (PVD) in a minimally invasive fashion, but mechanical injury to the diseased vessel is unavoidable. This damage results in endothelial cell disruption, exposure of the sub-endothelial matrix, and underlying vascular smooth muscle cells and initiates vascular wall remodeling that oftentimes contributes to the development of secondary vascular pathologies, such as intimal hyperplasia (IH)-induced restenosis [1,2]. Although there are approved therapeutic interventions available that are aimed at the inhibition of this dysfunctional remodeling (balloon-eluted pharmaceutics, drug-eluting stents, brachytherapy radiation, etc.) [3,4,5,6,7], there are currently no benchmark therapeutic options with proven long-term success. Spatially controlled nanoparticles designed to co-localize to exposed sub-endothelial matrices in injured vasculature could provide a mechanism for targeted vascular therapeutics, and the possibility of non-invasive interventions for both the acute and chronic phases of remodeling. 

Liposomal nanoparticles are promising drug delivery vehicles for potential translation due to their biocompatibility, relatively low cytotoxicity, and highly modifiable properties. Naturally occurring phospholipids can be arranged in bilayers that form spherical nanoparticles mimicking the cell membrane, and surface-linked polyethylene glycol (PEG) can be incorporated to increase membrane stability and pharmacokinetic profiles [8,9,10,11]. PEGylated liposomes (PLPs) are particularly promising in the field of targeted drug development due to their flexibility of modification and multifunctional potential, where PEG moieties provide an outer surface scaffold for the conjugation of functional ligands. Such modifications have allowed PLPs to be effectively functionalized for enhanced cellular uptake, cell-specific targeting, triggered release, imaging, and tissue localization [12,13,14,15]. 

We have previously described a PLP platform that is an effective delivery system in vascular cells types [16,17,18]. Our goal is to further develop these PLPs via surface modifications designed to preferentially target collagen type IV, a primary component of the vascular basement membrane exposed only in areas of endothelial perturbations. Here, we present the discovery-driven development of PLPs conjugated with short peptide fragments previously shown to bind collagen IV [19], and we describe the optimization of assembly and modification parameters for functional collagen-targeting PLP nanoparticles (CT-PLPs). 

## 2. Materials and Methods

### 2.1. Liposome Constituents

All liposome formulation constituents are defined in Table 1. Lipids and cholesterol were purchased from Avanti Polar Lipids (Alabaster, AL, USA). Collagen-targeting peptides (CTP) were custom purchased with an azido-modified lysine [KLWVLPKK(N_3_)-NH_2_] from P3 Biosystems (Louisville, KY, USA). 

### 2.2. CTP-Modified Lipid Synthesis

DSPE-PEG-DBCO, a form of DSPE-PEG with a cyclooctyne modification that can be used in azide-alkyne cycloaddition reactions, was used to form CTP-modified lipids (CTLs) via click chemistry reaction. Briefly, equilmolar azido-modified CTP peptides and DSPE-PEG-DBCO were combined at room temperature under constant agitation for 2 h, according to previously established reaction conditions [20,21]. The mass spectrometry confirmed peptide conjugation and reaction efficiency, whereby peaks of aggregate molecular weight were demonstrated in the reaction product spectra, compared to peaks in the CTP spectra and pre-clicked DSPE-PEG-DBCO spectra alone (Figure 1). 

### 2.3. Non-Targeting Liposome (PLP) Assembly

Base PLPs were formed with bulk lipid DOPC and Chol at a mole ratio of 7:3 plus 10 mol% DSPE-PEG and 0.5 mol% Rho-DPPE or 0.5 mol% Cy7-DOPE for fluorescent labeling. PLP nanoparticles were assembled using a previously described EtOH injection technique [16]. Briefly, lipids were dissolved in CHCl_3_, combined as indicated, and dried under N_2_ gas and vacuum to remove the remaining solvent. Dried lipid films were then resuspended in 100% molecular grade EtOH and injected dropwise into 10 mM Tris-HCl at pH 8.0 and a 2:3 EtOH:aqueous volume ratio, under constant vortexing at room temperature. Liposomes were purified from EtOH via 24 h dialysis against PBS at 4 °C and extruded using a 100 nm polycarbonate NanoSizerTM extruder prior to characterization (T&T Scientific, Knoxville, TN, USA). 

### 2.4. CT-PLP Assembly

CT-PLPs were formed by inserting 0.5–15 mol% CTL into base PLPs, either at lipid hydration (pre-insertion; PreCTL) or via micellar transfer (post-insertion; PostCTL). In all cases, DSPE-PEG was substituted for CTL at equal mol % in order to control the PEG-mediated membrane stability. 

#### 2.4.1. Modification via Pre-Insertion (PreCTL)

CTLs were combined with base PLP lipid constituents at the time of lipid drying under N_2_ gas. Lipid hydration with EtOH was performed in one step, and all lipid constituents were incorporated at the time of the initial liposome assembly. Liposomes were purified from EtOH via 24 h dialysis against PBS at 4 °C and extruded as described. 

#### 2.4.2. Modification via Post-Insertion (PostCTL)

Base PLPs were assembled as described, without the incorporation of CTL, and initially purified from EtOH via 2 h dialysis against PBS at 4 °C. CTLs were dissolved in CHCl_3_ and dried under N_2_ gas in a separate lipid film. Pre-formed base PLPs were combined with the CTL film and incubated at 37 °C under constant vortexing for 2 h to allow micellar transfer. Liposomes were further purified via 24 h dialysis against PBS at 4 °C and extruded as described. 

### 2.5. Liposome Characterization Studies

Figure 2 illustrates the assembled PLP and CT-PLP groups and their constituents in a schematic form. 

#### 2.5.1. Size and Homogeneity

The mean size, associated polydispersity index (PDI), and membrane z-potential of all liposome preparations were measured by dynamic light scattering (DLS) and relative electrophoretic mobility in water using the Zetasizer Nano ZS instrument (Malvern Instruments Ltd., Worchestershire, UK). 

#### 2.5.2. Morphologic Characterization by Scanning Transmission Electron Microscopy (STEM)

Liposome morphology and lamellarity were investigated by STEM using a negative-stain method. Liposomes were applied dropwise to a carbon film-coated copper grid and allowed to air dry. Liposome films were then stained with 2% phosphotungstic acid and air-dried for 1 min at room temperature. The samples were visualized with a Zeiss Auriga 40 STEM scope, and images were acquired by SmartSEM image acquisition software (Carl Zeiss, Inc., Oberkochen, Germany). 

### 2.6. Static Collagen-Binding Assays

Human collagen IV protein (Abcam, Cambridge, MA, USA) was reconstituted in 100 mM sodium acetate and diluted (*v*:*v*, 1:1) in 100% molecular grade EtOH to facilitate drying. Matrices were dried in 96-well plates at 3 µg/cm^2^ overnight at room temperature, washed in PBS and incubated with liposomes at 50 µM total lipid at 37 °C for 0–24 h. Unbound liposomes were rinsed 2× with PBS, and liposome binding was assayed in duplicate via the fluorometric detection of Rho-DPPE at 525 nm (GloMax Multi Reader, Promega, Madison, WI, USA). Static binding affinity was quantified as % lipid bound vs. 50 µM total lipid when compared to the fluorescence of an internal serial dilution curve of each relative liposome sample. Qualitative binding images were acquired by fluorescent microscopy using a Texas Red fluorescent filter at 400× under equivalent exposures across all groups (BX51 Olympus microscope, Olympus Q-color camera, Olympus Corporation, Shinjuku, Tokyo, Japan). 

### 2.7. Collagen Binding under Hemodynamic Flow

Human collagen IV protein was diluted as described and dried overnight at 3 µg/cm^2^ on a glass coverslip fitted to an FCS3 parallel-plate flow chamber (Bioptechs, Butler, PA, USA). Liposome binding under simulated hemodynamic flow was assayed using this parallel-plate chamber, within a closed circulation system where pulsatile flows and pressures were variably controlled via the chamber area and the pump speed (Figure 3). Liposomes at 50 nM total lipid were flowed across collagen matrices at 5–115 dynes/cm^2^ in order to mimic venous and arterial physiological to pathological flow conditions [22,23]. Dynamic real-time fluorescent microscopy was used to acquire images of CT-PLPs bound over time under all shear stress conditions. Real-time images were acquired at 100× in identical frames from 5 to 45 min, three independent collagen areas were defined for each image set, and those coordinates were applied to subsequent frames for the quantification of collagen binding, measured as the mean pixel intensity and calculated as % change over time. For each condition set, an internal non-collagen negative control area was used for normalization to the background fluorescence and to account for photobleaching over time. At 45 min of flow, the coverslips were removed and washed three times with PBS, images of each run were acquired at 100×, and the total CT-PLP and control PLP binding was quantified as the mean fluorescent intensity. Images were acquired using the described acquisition equipment, and all image quantification was performed using Image ProPremier software (Media Cybernetics, Inc., Rockville, MD, USA).

### 2.8. Human Vessel Explant Binding under Ex Vivo Perfusion

All procedures involving human specimens were carried out in accordance with the ethical standards of the institution and with the 1964 Helsinki Declaration. The study was reviewed by the UTGSM Institutional Review Board and granted a category #4 exemption (IRB #4297), because it involved only the collection of de-identified pathological specimens. Saphenous veins were procured from lower extremities amputated in an operating room at the University of Tennessee Medical Center as a standard of care and transported to the research laboratory to maintain tissue viability via bioreactor culture. Independent vessel segments were perfused for 30 min with 50 µM total lipid using a closed perfusion bioreactor system (Figure 4). Following perfusion, vessel explants were fluorescently imaged via a UVP iBox Studio In Vivo Imaging System (Analytik Jena US, LLC, Upland, CA, USA) with near-infrared (NIR) filters for the fluorometric detection of Cy7-DOPE. Binding was quantified as the mean intensity of bound Cy7-labeled lipid per vessel area, normalized to the background of non-perfused vessels using Vision Works software (Analytik Jena US). 

### 2.9. Statistical Analysis

All data were reported as mean ± SEM. Statistical analyses were performed using Student’s t-test or one-way ANOVA and a post-hoc Student–Newman–Keuls test using SigmaStat 3.5 software (Systat Software, Inc., San Jose, CA, USA). Probability (*p*) values of ≤0.05 were considered significant.

## 3. Results

### 3.1. EtOH Injection Assembly of CT-PLPs Results in Homogenous ~50 nm Liposome Samples

We have previously reported our EtOH injection technique to be an efficient method for assembling stable PLP nanoparticle populations of ~50–60 nm with enhanced cargo encapsulation [16]. Using this technique to incorporate CTL for CT-PLP assembly, liposome stability was confirmed at ≤5 mol% CTL modification. All CT-PLPs formed with ≤5 mol% CTL at pre-insertion demonstrated favorable nanoparticle sizes at or below ~50 nm with narrow size distributions (PDI) comparable to PLP controls (Table 2). Additionally, the incorporation of ≤5 mol% CTL demonstrated an inconsequential effect on membrane z-potential compared to PLP controls. The pre-insertion of 10 and 15 mol% CTL resulted in unfavorable nanoparticle characterization profiles, with increased particle size and aggregation and increased PDI approaching or above 0.40 (Table 2).

### 3.2. CT-PLPs Demonstrate Significant Binding Affinity for Collagen IV Matrices under Static Conditions Compared to Non-Targeted PLP Controls

CTL incorporation increased liposomal collagen binding in a dose-dependent manner (Figure 5). Specifically, the pre-insertion of 2.5 and 5 mol% CTL resulted in significant increases in binding over non-targeted PLP controls and all other CT-PLP groups after 2 h static incubation (2.5 mol%: 9.5% ± 1.1% lipid bound, 5 mol%: 11.0% ± 1.8% lipid bound, PLP: 1.7% ± 0.8% lipid bound; Figure 5, *n* = 3, * *p* < 0.05 vs. all other groups). Because the characterization profiles of 10 and 15 mol% CTL modification (Table 2) demonstrated nanoparticle aggregation and decreased homogeneity, we proposed the fluorescent signal of lipid bound after 2 h static incubation in these groups is a result of lipoplex formation and aggregation at the protein fibrils, not a result of significant nanoparticle binding affinity. 

### 3.3. CT-PLPs Formed by the Pre-Insertion of CTL Demonstrate Early and Increased Binding over CT-PLPs Formed by Post-Insertion and PLP Controls

When 5 mol% CTL was incorporated via both pre-insertion (PreCTL) and post-insertion (PostCTL) techniques, pre-insertion was determined to be the optimal modification strategy for CT-PLP assembly. PreCTL formulations demonstrated a significant increase in collagen binding compared to PLP controls as early as 15 min static incubation (14.6% ± 1.7% vs. 7.1% ± 0.8% lipid bound; Figure 6A, *n* = 3–4, * *p* < 0.05 vs. PLP). PreCTL binding continued to increase over time, reaching significance over PostCTL by 30 min (21.7% ± 2.2% vs. 13.8% ± 1.4% lipid bound; Figure 6A, *n* = 3–4, ** *p* < 0.05 vs. PLP and PostCTL). By 24 h total static incubation, only up to ~7% lipids were bound in PLP controls and ~14% lipids were bound in PostCTL formulations, whereas approximately 28.2% ± 3.6% total lipids were bound in the PreCTL formulation group (Figure 6A,B).

### 3.4. CT-PLPs Demonstrate Binding Affinity to Collagen IV Matrices under Flow Conditions Mimicking Vessel Wall Hemodynamics

Under a continuous flow, CT-PLPs increased binding over time to collagen IV matrices under simulated venous and arterial physiological and pathological conditions. When CT-PLPs flowed across matrices at 5 and 10 dynes/cm^2^, mimetic of venous physiological and pathological shear stress, respectively, liposome binding increased by 19% and 14% over 30 min, respectively (Figure 7A). At 45 min of the total flow, CT-PLPs at 10 dynes/cm^2^ demonstrated significant binding over PLP controls (6.2 ± 0.2 and 14.8 ± 0.6 AFU for PLP and CT-PLP, respectively; Figure 7B,C, *n* = 3, * *p* < 0.05 vs. PLP), while CT-PLP binding at 5 dynes/cm^2^ did not reach significance. Likewise, at shear stresses of 20 and 50 dynes/cm^2^ (mimetic of low and high physiological arterial flows), CT-PLP binding increase by 15% and 29% over 30 min, respectively (Figure 7A). At 45 min of the total flow, CT-PLPs also demonstrated significant binding over PLP controls at 20 dynes/cm^2^ (9.8 ± 0.1 and 22.0 ± 2.8 AFU for PLP and CT-PLP, respectively) and 50 dynes/cm^2^ (10.1 ± 0.9 and 33.1 ± 1.2 AFU for PLP and CT-PLP, respectively; Figure 7B,C, *n* = 3, * *p* < 0.05 vs. PLP). Even at a pathological shear stress of 115 dynes/cm^2^, CT-PLP binding increased by 22% over 30 min (Figure 7A) and bound significantly more compared to PLP controls (16.3 ± 2.9 and 43.7 ± 3.1 AFU for PLP and CT-PLP, respectively; Figure 7B,C, *n* = 3, * *p* < 0.05 vs. PLP). Ultimately, after a 45 min flow, CT-PLPs demonstrated remarkable binding under all conditions, while the non-targeted PLP control binding was negligible (Figure 7C).

### 3.5. CT-PLPs Demonstrate Binding Affinity to Human Lower Extremity Vessel Explants under Ex Vivo Vascular Perfusion

At 30 min of ex vivo liposomal vascular perfusion, CT-PLP binding to human saphenous vein segments was significantly enhanced over PLP binding (6.8 ± 2.0 and 19.9 ± 7.5 AFU for PLP and CT-PLP, respectively; Figure 8A,B, *n* = 3, * *p* < 0.05 vs. PLP). 

## 4. Discussion

Percutaneous interventions are minimally invasive, low-risk endovascular procedures for PVD with a majority of procedures, resulting in initial success rates of >90%. However, the thickening of the tunica intima is a common consequence of intervention-associated mechanical disruption. IH leads to clinical restenosis in an estimated 60% of PVD patients at 12-month post-percutaneous intervention, typically requiring additional endovascular procedures with increased morbidity and mortality rates [24,25]. During surgical and/or endovascular interventions, the mechanical disruption of the endothelial lining exposes sub-endothelial matrix proteins that could be leveraged for site-targeted or spatially controlled drug delivery, both at the time of intervention and by non-invasive administration during chronic remodeling.

Our lab and others have identified several molecular mechanisms that could be targeted for RNA interference (RNAi) therapy aimed at IH attenuation [26,27,28,29,30,31]. However, the development of molecular nanocarriers with translational potential will be required. Liposomal nanoparticles are considered advantageous for targeted drug delivery due to their biocompatibility, bio-degradability, low toxicity, low immunogenic response, improved pharmacokinetics, carrying capacity of both hydrophilic and hydrophobic drugs, and importantly their flexibility of modification. To this end, we have recently established a 3rd-generation liposome that incorporates PEGylation of naturally occurring lipids (i.e., PLPs), aimed at mitigating immunogenicity and improving stability, and peptide modifications that lend them as effective agents in the delivery of genetic material to vascular cell types and tissue [16]. Here, our objective was to leverage this technology to develop a surface-modified PLP platform functionalized for collagen-targeting (i.e., CT-PLPs).

Previous work by Chan et al. has established short peptide fragment KLWVLPKK as an effective ligand for preferential collagen IV binding properties [19]. Here, this collagen-targeting peptide was reacted with a natural PEGylated DOPE lipid using established click chemistry parameters to form collagen-targeting lipids. These lipids were then incorporated into our established PLP architecture to assemble our experimental CT-PLP formulations for functional optimization (Figure 1 and Figure 2). Initially, varying mol% CTL was tested to evaluate nanoparticle stability as a function of CTL incorporation and to determine an optimal modification level for functional binding when compared to our non-targeted PLP platform. This quality-by-design approach, to identify the most favorable characterization parameters early in formulation development, provides a rational step-by-step strategy focused on translational potential and future clinical application. Upon nanoparticle characterization, we demonstrated that CTL modification levels at or above 10 mol% resulted in liposome aggregation and loss of homogeneity (Table 2) and likely resulted in lipoplex aggregation at the collagen protein fibrils, instead of actual nanoparticle binding under static conditions (Figure 5). Interestingly, when screening the range of modification levels, we determined that CTL levels below 10 mol% were optimal for collagen binding under static conditions while still maintaining favorable critical quality attributes for translation (i.e., size, PDI, and z-potential). Specifically, when examining the binding capacity of CT-PLPs with the optimal characterization profiles, 2.5 and 5 mol% were the best performing CT-PLP groups, with significantly better collagen binding demonstrated above all other tested conditions (Figure 5). By maximizing the concentration of CTL surface-conjugated moieties, we proposed that we could optimize CT-PLP binding capacity under the hemodynamic complex system parameters that will be clinically required, so long as the maximal surface conjugation was not at the expense of imperative critical quality attributes. Therefore, while both 2.5 and 5 mol% CT-PLP demonstrated significant static binding, we chose to move forward with the higher concentration of 5 mol% CTL surface moieties for enhanced feasibility in a dynamic flow environment. 

Next, we aimed to determine the best technical approach to CTL incorporation and CT-PLP assembly. Common liposome modification techniques that can be used for peptide–lipid conjugate incorporation are as following: (1) the inclusion of amphiphiles during initial liposome assembly, known as pre-insertion; and (2) the insertion of amphiphiles into the bilayers of pre-formed liposomes via micellar transfer, known as post-insertion. Here, moving forward with our empirically derived 5 mol% CTL parameter, CT-PLPs assembled via the pre-insertion of CTL demonstrated significantly higher collagen binding affinity than that of PostCLT groups or PLP controls (Figure 6). Furthermore, the collagen binding of PreCLT particles occurred at earlier time points and continued to increase over time up to 60 min. This result was somewhat unexpected, as amphiphilic post-insertion is generally considered more energetically favorable with the hydrophobic lipid tail more readily inserted at the membrane interface, leaving the hydrophilic head and its peptide conjugate exposed to the outer surface for functionality [32,33,34]. With the pre-insertion technique, it could be postulated that a proportion of CTP is entrapped within the inner core of the particle, thereby losing some of the functional action of the peptide. Further investigation of post-insertion technique parameters may delineate better conditions for hydrophobic tail penetration, resulting in more effective assembly. Nevertheless, our results indicated the CT-PLP assembly by pre-insertion results in more efficacious collagen targeting and ultimately a more functional nanoparticle.

While these CT-PLPs were found to have a significant affinity to collagen matrices under static conditions, these empirically derived specifications and assembly parameters must imbue the resulting nanoparticles with a functional binding capacity that persists under dynamic flow conditions. Otherwise, the translational potential for targeted delivery in experimental in vivo and pre-clinical models is negligible. When collagen matrices were utilized in a closed circulation system where liposomes could be flowed across its surface under controlled pulsatile flow (Figure 3), CT-PLPs demonstrated notable binding. In fact, under conditions mimetic of true physiological and pathological shear stress in both the venous and arterial architectures, CT-PLPs bound effectively and in real time under a constant flow (Figure 7A). Furthermore, after 45 min of a continuous flow, CT-PLP binding was significant under all flow conditions, whereas the binding of non-targeted PLP controls was negligible (Figure 7B,C). Moreover, an unexpected result was the direct relationship between increased shear stress and increased CT-PLP binding affinity, exhibited by incremental increases in liposome binding from 5 to 115 dynes/cm^2^ (Figure 7B). We expected liposome binding to be less efficacious at higher stresses, but indeed there was an apparent increase in CT-PLP binding at shear stresses more mimetic of arterial flow and pathological flow. Targeted nanoparticle binding demonstrated under hemodynamic environments, particularly those mimetic of clinically relevant pathological conditions, supports the translational potential of this CT-PLP platform for both in vivo investigation and pre-clinical advancement. Finally, in order to demonstrate the feasibility of this nanocarrier formulation for clinical translation, CT-PLP binding was assayed in human vascular tissue explants under ex vivo vascular perfusion conditions (Figure 4). CT-PLP demonstrated significant binding to human saphenous vein explants compared to PLP controls (Figure 8), indicating significant potential for clinical development towards a translational vascular therapeutic. 

## 5. Conclusions

Here, we have established a collagen-targeting liposomal framework that shows promise in the development of a spatially controlled drug delivery platform. CT-PLPs herein described demonstrated an affinity to collagen IV binding under static conditions and under simulated hemodynamic flow ranging from physiological to pathological shear stresses. Furthermore, CT-PLPs demonstrated an affinity for binding human vascular tissue explants under ex vivo perfusion conditions. Ongoing studies aim to demonstrate CT-PLP binding capacity in situ via ex vivo venous and arterial perfusion under elevated shear stress in a pressure-controlled perfusion system. Moreover, we are working to demonstrate CT-PLP selective binding capacity to vascular targets via an in vivo rodent model of vascular injury and to determine the in vivo pharmacokinetic, biodistribution, and immunogenetic properties of CT-PLPs. These are instrumental steps in advancing this platform toward pre-clinical and clinical translation. 

Spatially controlled nanoparticles designed to co-localize to exposed sub-endothelial matrices could provide an optimal delivery system for targeted vascular therapeutics and theranostics. In future studies, using our in vivo pre-clinical rodent model of vascular injury, CT-PLP nanoparticles designed to selectively target the injury site in a vascular surgical model upon intravenous injection will be investigated for their potential therapeutic and theranostic use. By incorporating radio-labels or fluorescent labels within the technology to facilitate the tracking of nanocarriers, enabling localization capabilities within a complex system [35], we plan to assay CT-PLPs for their detection of areas of vascular perturbations via in vivo imaging. Furthermore, we have previously described various molecular mechanisms of vascular remodeling and cellular phenotypic differentiation that could be targeted for IH attenuation [26,27,28,29,30,31]. We intend to use utilize CT-PLPs for the delivery of RNAi aimed at these mechanisms to assay for their efficacious attenuation of vascular pathogenesis. Alternatively, CT-PLPs could be assayed for the targeted delivery of FDA-approved pharmaceutical compounds such as sirolimus and paclitaxel, as their utility in drug-eluting stents has recently fallen under scrutiny due to apparent systemic elution/distribution. Finally, by combining both imaging and therapeutic approaches, CT-PLPs could be utilized to deliver and demonstrate reparative therapy upon follow-up evaluation. Collectively, our long-term goal is to develop a spatially controlled nanocarrier for drug delivery at specific areas of vascular injury to provide a potential modality for targeted therapeutics and prevent post-intervention failure. 

## Figures and Tables

**Figure 1 pharmaceutics-13-01816-f001:**
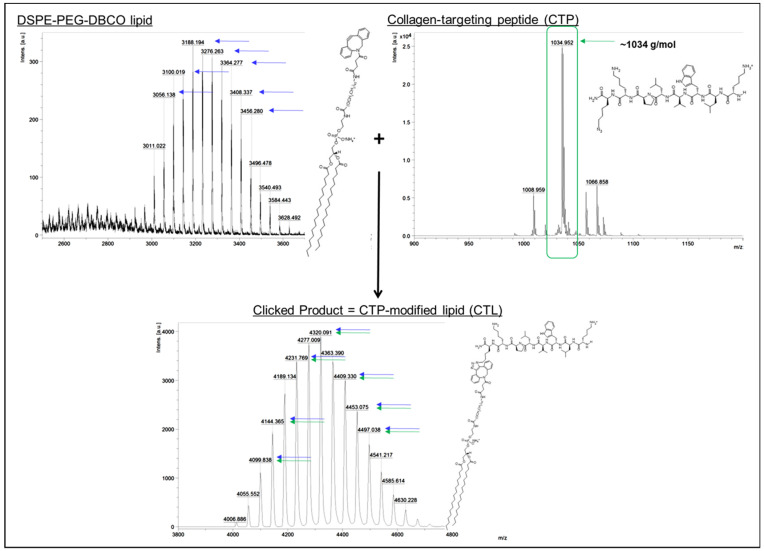
Spectra and structures of lipid/peptide constituents used for collagen-targeting peptide-modified lipid (CTL) conjugation. The mass spectrometry confirmed peptide conjugation, where peaks of aggregate molecular weight were demonstrated in the reaction product spectra, compared to peaks in the collagen-targeting peptide (CTP) spectra and pre-clicked DSPE-PEG-DBCO spectra alone.

**Figure 2 pharmaceutics-13-01816-f002:**
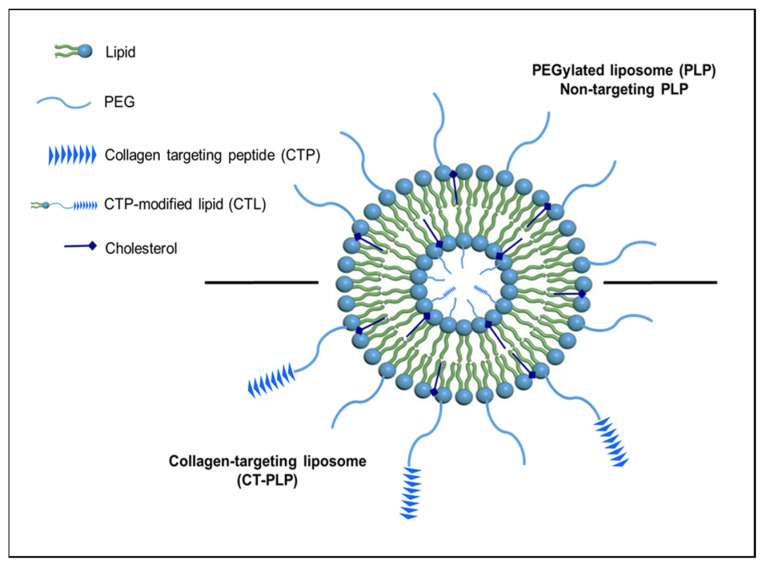
Schematic representation of collagen-targeting liposome (CT-PLP) and PEGylated liposome (PLP) controls. PLPs were assembled with bulk lipid DOPC and Chol at a mole ratio of 7:3 plus 10 mol% DSPE-PEG and 0.1 mol% rhodamine-DOPE for fluorescent labeling. CT-PLPs were formed likewise with the addition of CTP-conjugated CTL.

**Figure 3 pharmaceutics-13-01816-f003:**
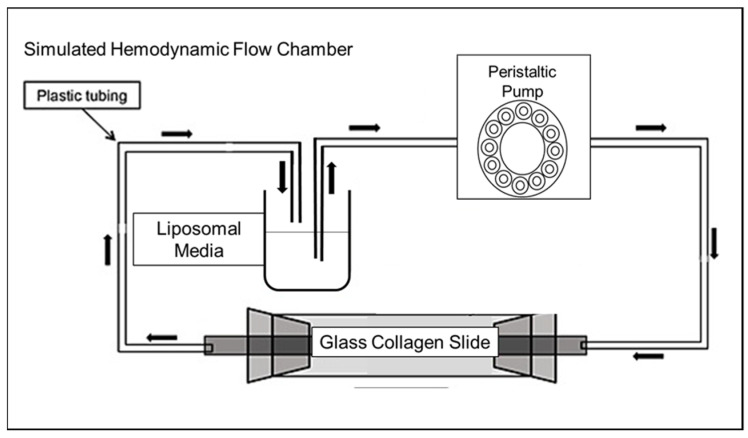
Schematic representation of the FCS3 parallel-plate flow chamber setup used for the simulated hemodynamic flow.

**Figure 4 pharmaceutics-13-01816-f004:**
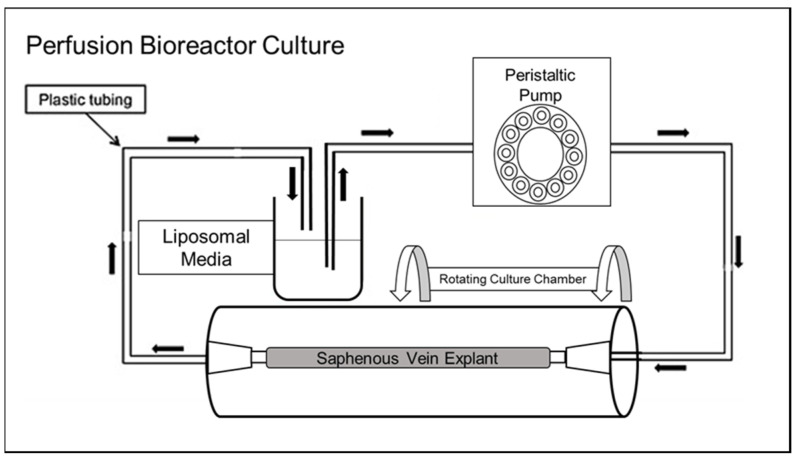
Schematic representation of the bioreactor setup used for human vessel explant perfusion.

**Figure 5 pharmaceutics-13-01816-f005:**
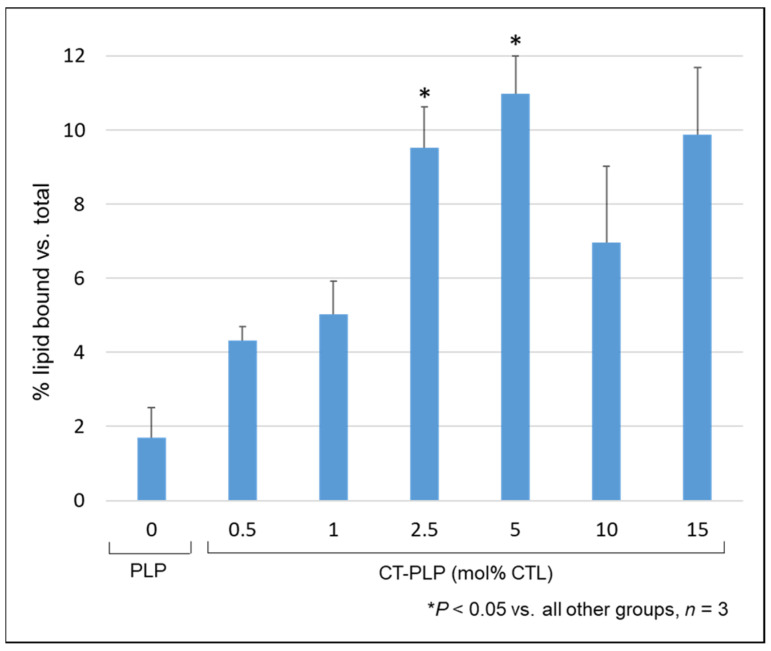
CTL incorporation increased liposomal collagen binding in a dose-dependent manner. The binding affinity of CT-PLPs assembled with 2.5 and 5 mol% CTL was significantly higher than in all other groups. Binding was detected by the fluorometry of Rho-labeled lipids, and data of the groups were presented as mean % lipid bound compared to respective total lipid serial dilution curves.

**Figure 6 pharmaceutics-13-01816-f006:**
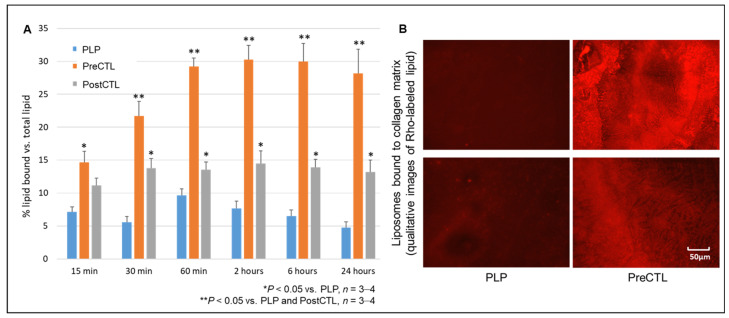
Pre-insertion was determined to be the optimal modification strategy for CT-PLP assembly. (**A**) When 5 mol% CTL was incorporated via pre-insertion (PreCTL), liposome binding was significantly increased over CTL incorporation via post-insertion (PostCTL) and over PLP controls at all time points. Binding was detected by the fluorometry of Rho-labeled lipids, and data in the groups were presented as mean % lipid bound compared to the respective total lipid serial dilution curves at each time point. (**B**) Representative fluorescent microscopy images showing Rho-labeled liposomes bound to collagen IV matrices after 24 h static incubation.

**Figure 7 pharmaceutics-13-01816-f007:**
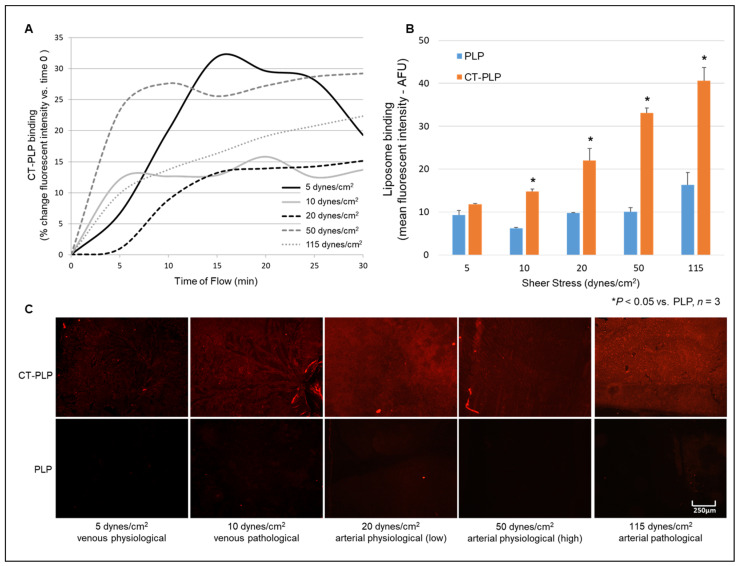
CT-PLPs bound to collagen IV matrices under a continuous flow. (**A**) At hemodynamic conditions, simulated at venous and arterial physiological and pathological sheer stress, CT-PLPs assembled with 5 mol% CTL via pre-insertion demonstrated increased binding over time. (**B**) After 45 min of a continuous flow, CT-PLP binding was significantly increased over PLP control binding at all simulated venous and arterial physiological and pathological sheer stresses. Binding was detected by the fluorescent microscopy of Rho-labeled lipids, and data were presented as the mean fluorescent intensity normalized to the background fluorescent intensity. (**C**) Representative fluorescent microscopy images showing Rho-labeled CT-PLP and PLP liposomes bound to collagen IV matrices after 45 min of a continuous flow at the simulated venous and arterial physiological and pathological sheer stresses.

**Figure 8 pharmaceutics-13-01816-f008:**
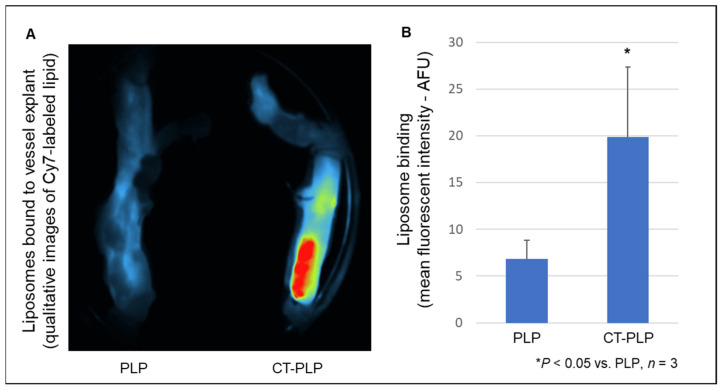
CT-PLP binding was significantly increased over PLP controls in human lower extremity vessel explants under ex vivo vascular perfusion. (**A**) Representative near-infrared (NIR) images showing Cy7-labeled liposomes bound to saphenous vein segments after 30 min continuous ex vivo perfusion. (**B**) After 30 min of a continuous flow, CT-PLP binding was significantly increased over PLP controls. Binding was detected using in vivo imaging system software to quantify fluorescent intensity per vessel area, and data were presented as the mean fluorescent intensity normalized to the fluorescent intensity of the background of non-perfused control vessels.

**Table 1 pharmaceutics-13-01816-t001:** Defined liposome constituent acronyms.

Lipid Derivative/Constituent	Acronym
1,2-distearoyl-sn-glycero-3-phosphoethanolamine-*N*-[methoxy(polyethylene glycol)-2000]	DSPE-PEG
1,2-distearoyl-sn-glycero-3-phosphoethanolamine-*N*-[dibenzocyclooctyl(polyethylene glycol)-2000]	DSPE-PEG-DBCO
1,2-dioleoyl-sn-glycero-3-phosphocholine	DOPC
*N*-(lissamine rhodamine B sulfonyl)-1,2-dipalmitoyl-sn-glycero-3-phosphoethanolamine	Rho-DPPE
*N*-(Cyanine 7)-1,2-dioleoyl-sn-glycero-3-phosphoethanolamine	Cy7-DOPE
Ovine cholesterol KLWVLPKK(N_3_)-NH_2_	Chol CTP

**Table 2 pharmaceutics-13-01816-t002:** Liposome characterization profiles.

	**Mol % CTL**	**Size (nm)**	**PDI**	**z-Potential**
PLP	0	60 ± 9	0.21 ± 0.07	6.20 ± 0.75
CT-PLP	0.5	46 ± 2	0.32 ± 0.20	6.03 ± 1.28
	1	49 ± 2	0.12 ± 0.01	7.62 ± 0.22
	2.5	50 ± 2	0.13 ± 0.01	7.35 ± 0.43
	5	51 ± 3	0.17 ± 0.01	8.03 ± 0.27
	10	508 ± 210	0.44 ± 0.05	8.02 ± 0.17
	15	588 ± 158	0.38 ± 0.10	9.01 ± 0.29

## Data Availability

Data are contained within the article.

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
