# Peer review of "Liposomal Nanocarriers Designed for Sub-Endothelial Matrix Targeting under Vascular Flow Conditions"

_pharmaceutics, 2021, doi:10.3390/pharmaceutics13111816_

Round 1
Reviewer 1 Report
The paper under consideration will definitely be interesting for the readers of Pharmaceutics. It is well-planned and accurate research paper devoted to targeted liposomal delivery.
I have only a few points to note:
- Please correct indexes in chemical formulas in 2.3 section
- Concerning Fig. 5: Do the authors propose any ideas about significant decrease of binding for 10% mol, while for 15% this value is higher?
- Is the binding/capture of such liposomes by immune system cells expected despite PEGylation?
Reviewer 2 Report
I believe that the manuscript proposed is very well written and demonstrated clearly that the addition of the collagen binding peptide dramatically increase the retention of the proposed liposomes to the vasculature.
I suggest a minor revision. The following suggestions are not mandatory, but they could help the reader in understanding the results.
Main concern: In the discussion, you mentioned that these liposomes could be used to deliver therapeutic molecules or imaging agents. However, it is not clear what is your actual future plan for CT-LPLs. Which molecules would you like to include in future formulations? I believe that your ‘’plan’’ should be specified somewhere. If not CT-PLP seems simply empty liposomes, at the moment, with no therapeutic (or imaging) capacities. See also comment 7 (it seems somewhat concerning to me)
1) I suggest to combine Figure 5 with Figure 6 and Figure 7 with Figure 8. Furthermore, graphs in the Figures looks quite big and their size could be reduced (it is not necessary to include the figure number in the figures but I believe it is sufficient to specify it in the figure legends).
2) It is not specified in the text the Z potential of PLP and CT-PLP. If you have this data, I recommend to add it to table 2. It is not clear if the added peptide changes the overall charge of the liposomes. This could be particularly relevant if you would like to use it for delivery of nucleic acids (since the formulation includes DOPE).
3) Please specify in Figure 8a what are the numbers represented on the X axis (In the figure or in the figure legend). In Figure 8B, it is not specified what is represented in the images in the first-row vs the second row (Top CT-PLP, bottom PLP?). Please specify this in the figure legend. Furthermore, I suggest to delete the first phrase of the figure legend (which summarizes the results) and maybe explain the details of the experiments a bit more in depth (it is hard to comprehend for someone who doesn’t work on vascular targeting).
4) Please specify in the text why you have decided to use 5mol% CTL instead of 2.5mol%. In Figure 5, you showed no significant differences between 2.5 and 5 so it is not clear why you have chosen 5% for further experiments. In addition, it is not discussed why the addition of the peptide lowers the overall size of the PLP (from 60 to 50). Is this not significant? It looks significant to me.
5) It could be better to change the numbering of the figures. Figure 2 could become the graphical abstract while Figure 5 should be regarded as Figure 1 (since it is the first figure showed in the results section)
6) In the discussion (line 334) you wrote that liposomes have low immunogenicity. However, it is known that pegylation could results in the formation of anti-peg antibodies which could limit their effectiveness after multiple injections. Maybe this could be relevant for in vivo studies in the future and could be discussed if you plan multiple injections in the future.
7) It is not clear if the vessels used for the experiments in figure 9 are simply normal vasculature or are somewhat damaged. You previously specify that collagen IV is exposed in damaged vessels. I believe could be worth expanding on this point in the text or maybe show that these vessels have collagen IV on their surface? It seems that CT-LPL will interact with all the vessels in the body (if these vessels are not damaged somehow?) So how CT-LPL could spare from damaged vs healthy vessels?
